# Genetic Modulation of Silodosin Exposure and Efficacy: The Role of CYP3A4, CYP3A5, and UGT2B7 Polymorphisms in Benign Prostatic Hyperplasia Management

**DOI:** 10.3390/jpm15080386

**Published:** 2025-08-18

**Authors:** Shokhrukh P. Abdullaev, Maksim N. Shatokhin, Pavel O. Bochkov, Svetlana N. Tuchkova, Oleg B. Loran, Sherzod P. Abdullaev, Karin B. Mirzaev, Dmitry A. Sychev

**Affiliations:** 1Federal State Budgetary Educational Institution of Further Professional Education “Russian Medical Academy of Continuous Professional Education”, Ministry of Healthcare of the Russian Federation, Barrikadnaya Str. 2/1, Bld. 1, Moscow 125993, Russia; luon@mail.ru (S.P.A.); sh.77@mail.ru (M.N.S.); bok-of@yandex.ru (P.O.B.); tuchkovasn@rmapo.ru (S.N.T.); loranob@rmapo.ru (O.B.L.); karin05doc@yandex.ru (K.B.M.); dimasychev@mail.ru (D.A.S.); 2Private Healthcare Institution “Central Clinical Hospital ‘RZD-Medicine’”, Volokolamsk Highway 84, Moscow 125367, Russia

**Keywords:** silodosin, pharmacokinetics, polymorphism, UGT2B7, CYP3A4, CYP3A5, ABCB1, benign prostatic hyperplasia, pharmacogenetics

## Abstract

**Objectives**: Silodosin, a selective α1A-adrenoceptor antagonist, is used to treat lower urinary tract symptoms (LUTS) associated with benign prostatic hyperplasia (BPH). Genetic polymorphisms in drug-metabolizing enzymes and transporters may contribute to interindividual variability in its efficacy and safety. This study aimed to investigate the influence of CYP3A4, CYP3A5, UGT2B7, and ABCB1 polymorphisms on silodosin pharmacokinetics, efficacy, and safety in Russian patients with BPH. **Methods**: A prospective observational study included 103 Russian male patients with moderate-to-severe LUTS (IPSS > 8) due to BPH, treated with silodosin (8 mg daily) for 8 weeks. Genotyping for CYP3A4*1B, CYP3A4*22, CYP3A5*3, UGT2B7 (rs73823859, rs7439366, and rs7668282), and ABCB1 (rs4148738, rs1045642, rs2032582, and rs1128503) was performed using real-time PCR. The silodosin minimum steady-state plasma concentration (Css min) was measured via HPLC-MS. Efficacy was evaluated by the International Prostate Symptom Score (IPSS), quality of life scale, maximum urinary flow rate (Qmax), residual urine volume (RUV), and prostate volume at the baseline and week 8. Adverse drug reactions (ADRs) were recorded. **Results**: CYP3A4*22 CT carriers (n = 6) exhibited higher Css min (17.59 ± 2.98 vs. 9.0 ± 10.47 ng/mL, *p* = 0.049) but less absolute IPSS improvement (*p* < 0.05), likely due to higher baseline symptom severity. However, the change in IPSS (ΔIPSS_1–4_) from the baseline to week 8 did not differ significantly (−5.78 ± 5.29 vs. −6.0 ± 4.54, *p* = 0.939). CYP3A5*3 GG homozygotes (n = 96) showed greater ΔIPSS_1–4_ improvement (−6.25 ± 4.60 vs. 0.0 ± 9.53, *p* = 0.042) and a lower IPSS at day 28 (7.64 ± 4.50 vs. 20.0 ± 6.55, *p* < 0.001). UGT2B7 rs7439366 TT carriers (n = 34) had an improved Qmax (ΔQmax_1–4_ 5.4 vs. 3.3 and 2.0 mL/s for CC and CT, *p* = 0.041). ABCB1 1236C>T TT homozygotes (n = 25) showed a trend toward reduced RUV (*p* = 0.053). No polymorphisms were associated with adverse drug reactions (15 events in 42 patients, 35.7%). **Conclusions**: Genetic polymorphisms CYP3A4*22, CYP3A5*3, and UGT2B7 rs7439366 may modulate silodosin pharmacokinetics and efficacy parameters in BPH patients but not safety. Larger-scale studies are warranted to validate these initial findings.

## 1. Introduction

Benign prostatic hyperplasia (BPH) is a non-malignant condition of the prostate characterized histologically by an increase in glandular epithelial tissue, smooth muscle, and connective tissue in the transitional zone of the prostate [1,2]. A meta-analysis incorporating data from 25 countries reported a lifetime prevalence of BPH of 26.2% (95% CI: 22.8–29.6%) [3]. Projections indicate a global increase in benign prostatic hyperplasia (BPH) incidence and prevalence, rising from approximately 962 and 7879 per 100,000 people in 2022 to about 999 and 8621 per 100,000 by 2035 [4]. Based on 2019 US Medicare data, global healthcare costs for benign prostatic hyperplasia (BPH) are estimated to be approximately USD 73.8 billion annually, as extrapolated from US expenditure trends [5].

BPH is the primary cause of lower urinary tract symptoms (LUTS), which include frequent or urgent urination, a weak urinary stream, difficulty initiating urination, nocturia (nighttime urination), and incomplete bladder emptying. If left untreated, BPH can lead to chronic urinary retention with elevated intravesical pressure—a potentially life-threatening condition—and long-term or irreversible changes in the detrusor muscle of the bladder. A systematic review of 222 studies across 36 countries reported that approximately 63% of men experience LUTS associated with BPH, with about 31% having moderate-to-severe symptoms (95% CI: 58.0–68.1 and 28.8–33.8, respectively). The prevalence of LUTS has risen over time, from roughly 27.4% in the 1990s to 31.9% in the 2000s and 36.2% in the 2010s (95% CI: 24.5–30.3, 27.3–36.7, and 30.7–41.9, respectively) [6].

Current clinical guidelines from the European Association of Urology and the American Urological Association recommend selective α1-adrenoreceptor blockers (ABs) as a first-line therapy for alleviating LUTS associated with BPH [7,8]. Tamsulosin and silodosin are preferred over other ABs for BPH treatment, primarily due to their high selectivity for the α1A-adrenoreceptor subtype, which is predominantly located in the prostate and urethra. This urospecificity results in the effective relief of LUTS with fewer systemic adverse effects associated with the blockade of α1B and α1D adrenoreceptors, such as orthostatic hypotension. However, 30–40% of patients experience suboptimal symptom improvement, defined as a reduction in the International Prostate Symptom Score (IPSS) of less than 30%, or an improvement of fewer than three points on the IPSS scale following treatment initiation [4,9,10]. Factors contributing to the suboptimal response include the following: a high baseline severity of LUTS (IPSS > 20), which reduces the likelihood of significant improvement; a larger prostate volume (>40 mL), which often correlates with less symptom improvement with ABs alone; and elderly age or comorbidities, which may influence symptom perception [11]. Safety concerns with ABs also remain a challenge. Common adverse drug reactions (ADRs) associated with tamsulosin and silodosin include dizziness, orthostatic hypotension, ejaculatory dysfunction, and asthenia. This class of drugs is associated with a modest but significant increase in the risk of falls (odds ratio [OR]: 1.14; 95% CI: 1.07–1.21) and fractures (OR: 1.16; 95% CI: 1.04–1.29) [12]. The higher selectivity of silodosin for α1A-adrenoreceptors in the lower urinary tract reduces the risk of adverse cardiovascular effects but increases the risk of ejaculatory dysfunction [13,14]. Age, comorbidities, and concomitant therapies may further influence the safety profile of ABs. The frequency and severity of ADRs vary among individuals, likely due to differences in receptor subtype distribution and variability in drug pharmacokinetics.

Another factor contributing to the variability in the efficacy and safety of ABs is the genetic profile of patients. Although pharmacogenomic studies analyzing genetic polymorphisms affecting the safety profile of tamsulosin and silodosin are limited [15,16,17,18], it is evident that variability in drug-metabolizing enzymes, such as the cytochrome P450 (CYP) system, contributes to interindividual differences in pharmacological response. The metabolism of ABs is mediated by the enzymes CYP3A4 and CYP2D6, and their genetic polymorphic variants can influence plasma drug levels and, consequently, the risk of toxicity. The CYP3A enzyme family exhibits broad substrate specificity and is involved in the metabolism of over 50% of pharmaceutical drugs [19]. CYP3A4 is considered the dominant isoform within the CYP3A family. Silodosin undergoes metabolism via direct glucuronide conjugation mediated by UDP-glucuronosyltransferase 2B7 (UGT2B7), with CYP3A4 catalyzing its oxidative reactions. In contrast, tamsulosin is primarily metabolized by CYP3A4 and CYP2D6 [20].

Our previous studies involving patients with LUTS/BPH treated with tamsulosin demonstrated that the carriage of CYP2D6 allelic variants, specifically *10 and *41, is associated with a greater improvement in LUTS and a reduction in residual urine volume. Patients classified as “CYP2D6 intermediate metabolizers” exhibited a better therapeutic response [21,22]. Carriers of the CYP2D6 *10/*10 or *5/*10 genotypes showed higher plasma concentrations of tamsulosin, which may enhance the drug’s efficacy but is also associated with a more pronounced reduction in blood pressure [15]. In contrast, CYP3A4 and CYP3A5 variants did not influence the efficacy or safety outcomes of tamsulosin therapy in BPH patients [23]. No similar studies have been conducted for silodosin in patients with LUTS due to BPH.

Thus, the aim of this study was to investigate the influence of polymorphisms in the drug-metabolizing enzyme genes CYP3A4, CYP3A5, and UGT2B7, as well as the P-glycoprotein transporter gene (ABCB1), on the efficacy, safety, and pharmacokinetics of silodosin in patients with LUTS/BPH.

## 2. Materials and Methods

The study was conducted in accordance with the Declaration of Helsinki and approved by the Local Ethics Committee of the Federal State Budgetary Educational Institution of Further Professional Education “Russian Medical Academy of Continuous Professional Education” of the Ministry of Healthcare of the Russian Federation (RMACPE) (Protocol No. 13, dated 27 December 2021).

Study Design: This was a prospective observational study conducted from 15 October 2023 to 30 May 2025. The study included patients receiving outpatient treatment at the Department of Endoscopic Urology of the RMACPE. Prior to enrollment, each patient provided written informed consent (IC) for participation in the study.

All participants provided written IC, which included permission to use anonymized data for scientific publications. The present work contains no personal data that could allow for the identification of patients.

Inclusion Criteria were as follows: signed IC; a confirmed diagnosis of BPH (ICD-10 code: N40); complaints of moderate-to-severe LUTS, as assessed by an IPSS > 8 points; and a prostate-specific antigen level < 4 ng/mL.

Non-inclusion Criteria included the following: complicated course of BPH; any conditions other than BPH that, in the doctor’s opinion, could cause dysuria or alter urinary flow rate (e.g., neurogenic bladder, bladder neck sclerosis, urethral stricture, acute or chronic prostatitis, and acute or chronic urinary tract infections); concurrent oncological diseases; severe concomitant cardiovascular conditions (e.g., unstable angina, recent myocardial infarction, or poorly controlled arterial hypertension) and cerebrovascular diseases (e.g., recent stroke or spinal cord injuries); and renal or hepatic insufficiency.

Exclusion criteria were as follows: the presence of any contraindications to the use of silodosin; patient refusal to adhere to the prescribed pharmacotherapy; and patient withdrawal of consent to participate in the study.

The study included 103 male patients with complaints of LUTS and a confirmed diagnosis of BPH. Patients were prescribed silodosin at a dose of 8 mg daily, as indicated. Treatment with silodosin commenced at the time of the patient’s initial visit, with follow-up continuing for at least 8 weeks. Treatment was prescribed in accordance with national clinical guidelines for the management of patients with BPH.

On the day of enrollment, 4 mL of blood was collected from each patient into sterile disposable vacuum tubes containing EDTA for subsequent genotyping. Blood sampling was performed concurrently with routine laboratory tests, eliminating the need for additional venipunctures. The biological material was frozen at −20 °C, transported to the laboratory, and subsequently stored at −70 °C.

The laboratory component of the study was conducted at the Research Institute of Molecular and Personalized Medicine of the RMACPE. For all patients enrolled in the study, a real-time polymerase chain reaction was used to determine the carriage of polymorphic markers CYP3A4*1B (c.-392G>A, rs2740574), CYP3A4*22 (c.522-191C>T, rs35599367), CYP3A5*3 (A6986G, rs776746), UGT2B7 (rs73823859, rs7439366, and rs7668282), and ABCB1 (rs4148738, rs1045642, rs2032582, and rs1128503).

For all patients, the minimum steady-state plasma concentration (Css min) of silodosin was determined. Plasma samples were collected in the morning immediately before the administration of the next daily dose, and no earlier than 5 days after the initiation of silodosin treatment to ensure steady-state conditions. The Css min of silodosin was measured using high-performance liquid chromatography on an Agilent 1200 liquid chromatograph (Agilent Technologies Inc., Santa Clara, CA, USA, 2008). Sample preparation was performed using the protein precipitation method. Detection of silodosin spectra was conducted using an Agilent Triple Quad LC/MS 6410 mass spectrometer with electrospray ionization in positive ion mode.

The efficacy of silodosin therapy was evaluated using a combination of clinical and instrumental methods. Clinical assessment included the evaluation of LUTS using the IPSS questionnaire. Instrumental assessments comprised the measurement of maximum urinary flow rate (Qmax), residual urine volume (RUV), and prostate volume (PV) determined by ultrasound. IPSS evaluations were conducted on days 1, 14, 28, and 56 of treatment and follow-up. Instrumental assessments of treatment efficacy were performed on days 1 and 56 of silodosin therapy.

Statistical Analysis: Data were analyzed using parametric and non-parametric statistical methods with the STATISTICA v10.0 software package (StatSoft Inc., Tulsa, OK, USA). The choice of statistical method was based on the normality of data distribution, which was assessed using the Shapiro–Wilk W-test and the Kolmogorov–Smirnov test. For non-normally distributed parameters, descriptive statistics were reported as the median (Me) and interquartile range (25th and 75th percentiles, Q1 and Q3). For normally distributed parameters, descriptive statistics were presented as the mean (M) with standard deviation (SD). Normally distributed data were compared using a Student’s *t*-test for paired or unpaired groups. Frequency characteristics of categorical variables were compared using a Pearson’s χ^2^ test. Comparisons between two independent groups were performed using the Mann–Whitney U test, while three or more groups were compared using the Kruskal–Wallis rank analysis of variance, followed by pairwise comparisons with the Mann–Whitney U test and Bonferroni correction for *p*-value adjustment. A significance level of *p* < 0.05 was considered statistically significant.

## 3. Results

A total of 103 male patients with BPH and LUTS were included in the study (median age: 69.5 years [64; 76], BMI: 28.72 kg/m^2^ [26.99; 31.49]). The majority of patients had comorbidities, with cardiovascular conditions present in 63.1% (n = 65) of the cohort, including hypertension (38.83%, n = 40) and ischemic heart disease (19.42%, n = 20). Additionally, type 2 diabetes mellitus was observed in 6.8% (n = 7), and urological conditions in 9.71% (n = 10) of patients. Common concomitant therapies included CYP3A4 substrates (amlodipine, atorvastatin, losartan, and nifedipine). Detailed clinical and demographic characteristics are presented in Table 1.

The genotype distribution of the investigated polymorphisms conformed to the Hardy–Weinberg equilibrium (*p* > 0.05), except for ABCB1 1236C>T (rs1128503, *p* = 0.018). A high frequency of the CYP3A5*3 allele (G, 96.1%) was observed, consistent with patterns in European populations. The CYP3A4*1B (rs2740574, A>G) and CYP3A4*22 (rs35599367, C>T) variants were rare, with minor allele frequencies (MAF) of 1.9% and 2.9%, respectively. The complete genotype distribution of the studied polymorphisms is presented in Table 2.

The analysis revealed that carriers of the CYP3A4*22 CT genotype (n = 6) exhibited a significantly higher Css min of silodosin compared to CC homozygotes (n = 97) (17.59 ± 2.98 ng/mL vs. 9.0 ± 10.47 ng/mL, *p* = 0.049). However, CT genotype carriers showed a less pronounced improvement in IPSS at all evaluation timepoints (days 14, 28, and 56) versus CC homozygotes (n = 36) (*p* < 0.05 for all visits). Baseline IPSSs were comparable between the CC and CT groups (18.5 ± 7.18 vs. 14.21 ± 4.94, *p* = 0.065). Notably, the overall change in IPSS (ΔIPSS_1–4_) and quality of life (ΔQoL_1–4_) from baseline to day 56 did not differ significantly between groups (*p* = 0.939 and *p* = 0.520, respectively).

The silodosin trough Css min showed no significant association with polymorphisms in CYP3A4*1B (c.-392G>A, rs2740574), CYP3A5*3 (c.6986A>G, rs776746), UGT2B7 (rs73823859, rs7439366, and rs7668282), ABCB1 (rs4148738, rs1045642, rs2032582, and rs1128503), and ABCG2 (rs2231142, 421C>A) (all *p* > 0.05; see Table 3).

Patients with the CYP3A5*3 GG genotype (n = 96) showed significantly greater improvement in IPSSs by treatment day 56 (ΔIPSS_1–4_ = −6.25±4.60) compared to those carrying the A-allele (AA + AG genotypes, n = 7; ΔIPSS_1–4_ = 0.0 ± 9.53, and *p* = 0.042). At the day 28 assessment, GG carriers maintained superior clinical outcomes (IPSS 7.64 ± 4.50 versus 20.0 ± 6.55 in A-allele carriers; *p* < 0.001). QoL measures did not differ significantly between groups (*p* > 0.05).

No statistically significant differences were observed in IPSS and QoL efficacy outcomes among patients with different ABCB1 (P-glycoprotein) genotypes. However, for the rs4148738 C>T polymorphism, a trend toward greater IPSS improvement was noted in TT homozygotes (n = 16, ΔIPSS_1–4_ = −9.16 ± 6.43) compared to CT heterozygotes (n = 59, ΔIPSS_1–4_ = −4.62 ± 4.55) and CC homozygotes (n = 28, ΔIPSS_1–4_ = −6.5 ± 5.28), though these differences did not reach statistical significance (*p* = 0.135).

Analysis of genetic polymorphisms’ effects on urodynamic parameters (Table 4) revealed limited but statistically significant associations. Patients with the UGT2B7 (rs7439366) TT genotype (n = 34) demonstrated a significantly greater improvement in the maximum urinary flow rate (Qmax) by day 56 of treatment compared to CC and CT genotypes (median Qmax: 12.75 mL/s [10.1;14.0] vs. 12.5 [10.1;27.6] and 11.0 [0.0;30.0], respectively; *p* = 0.043). The ΔQmax_1–4_ change in TT genotype carriers was 5.4 mL/s [0.9;9.6] versus 3.3 [−0.8;11.0] in CC and 2.0 [0.7;3.0;] in CT carriers (*p* = 0.041).

For the ABCB1 1236C>T variant, a trend toward reduced residual urine volume (ΔRUV_1–4_) was observed in TT homozygotes (n = 25): 1.0 [−10.0;5.0] mL versus −0.5 [−7.15;7.5] in CC and 3.0 [−10.0;14.0] in CT genotypes (*p* = 0.053). The polymorphisms CYP3A4*1B, CYP3A4*22, CYP3A5*3, UGT2B7 (rs73823859, rs7668282), and ABCB1 (rs4148738, rs1045642, and rs2032582) showed no significant effect on RUV, Qmax, or PV dynamics (*p* > 0.05 for all comparisons). The absence of PV changes across all groups (median ΔPV_1–4_ range: −1.0 to +6.5 cm^3^) confirms that silodosin does not affect the morphological component of BPH.

Interestingly, for CYP3A4*22 (CT genotype)—despite its association with differences in Css min and clinical response—no correlation was found with urodynamic parameters (*p* = 0.830 for ΔRUV_1–4_ and *p* = 0.492 for ΔQmax_1–4_, respectively).

During the observation period, 15 ADRs were recorded in 42 patients (35.7%). ADRs included the following: rhinitis (n = 5, 33.3%), orthostatic hypotension (n = 4, 26.6%), retrograde ejaculation (n = 2, 13.3%), dizziness (n = 2, 13.3%), diarrhea (n = 1, 6.6%), and headache (n = 1, 6.6%). Analysis of polymorphisms’ association with ADR incidence revealed no statistically significant correlations for any genetic variants (*p* > 0.05 for all). Odds ratios for all analyzed polymorphisms were non-significant.

## 4. Discussion

This study represents the first comprehensive investigation examining the effects of polymorphisms in CYP3A4*1B (c.-392G>A, rs2740574), CYP3A4*22 (c.522-191C>T, rs35599367), CYP3A5*3 (c.6986A>G, rs776746), UGT2B7 (rs73823859, rs7439366, and rs7668282), and ABCB1 (rs4148738, rs1045642, rs2032582, and rs1128503) on both the pharmacokinetics and clinical effects of silodosin in BPH patients. The key findings demonstrate that genetic variants of CYP3A4, CYP3A5, and UGT2B7 may contribute to the modulation of pharmacokinetic profiles and therapeutic responses, though they showed no association with drug safety.

Silodosin undergoes extensive metabolism primarily via CYP3A4, UDP-glucuronosyltransferase 2B7, and alcohol/aldehyde dehydrogenases. CYP3A4 catalyzes its oxidative metabolism, producing hydroxylated and ketonized derivatives. Predominantly expressed in the liver and intestine, CYP3A4 significantly influences oral bioavailability, systemic clearance, and consequently, the drug’s systemic exposure. CYP3A4 activity exhibits substantial interindividual variability due to genetic polymorphisms, environmental factors, health status, and comorbidities. The CYP3A4*18B and CYP3A4*22 variants demonstrate particularly pronounced effects on drug metabolism. While CYP3A4*18B reduces the metabolism of some drugs (e.g., requiring lower fentanyl doses for postoperative analgesia in *18B/*18B homozygotes [24]), it increases the clearance of others like cyclosporine and tacrolimus [25,26], reflecting the substrate-dependent modulation of enzyme activity. This variant predominates in East Asian populations (Chinese, Korean, and Japanese) [27,28] but is virtually absent in Caucasians. The CYP3A4*22 variant (rs35599367 C>T), located in intron 6, causes alternative splicing that reduces CYP3A4 expression and activity by 40–50% [29]. With a minor allele frequency of ~5% in Caucasians versus < 0.6% in Asians [30], CYP3A4*22 alters the clearance and pharmacologic responses of various CYP3A4 substrates—reducing quetiapine [31] and tacrolimus [32] dosing requirements while increasing cyclosporine exposure and toxicity risk [33].

In our study, carriers of the CYP3A4*22 allele (CT genotype) showed 95% higher trough silodosin concentrations (Css min: 17.59 vs. 9.0 ng/mL, *p* = 0.049), consistent with this variant’s known mechanism. As silodosin is a CYP3A4 substrate, reduced metabolic activity predictably increases systemic exposure. Paradoxically, despite higher concentrations, CT genotype patients demonstrated less symptom improvement by absolute IPSSs (*p* < 0.05 at all visits). However, the overall IPSS reduction by day 56 was comparable between the CC and CT groups (ΔIPSS_1–4_: −5.78 ± 5.29 vs. −6.0 ± 4.54, *p* = 0.939), potentially explained by the CT group’s more severe baseline symptoms (baseline IPSS: 18.5 vs. 14.21). Similarly, QoL assessments showed significant differences at all visits (*p* < 0.05), but the ΔQoL_1–4_ was comparable between the CC and CT groups (*p* = 0.520). LUTS reflect both dynamic obstruction (responsive to α1-blockers) and mechanical obstruction from prostate volume or detrusor changes (less responsive to α1-blockers). Our data show no significant prostate volume differences between the CC and CT groups (38.75 vs. 36.6 cm^3^, *p* = 0.86). Further large-scale studies are necessary to fully assess polymorphism effects on clinical outcomes, especially concerning prostate volume.

The lack of correlation between CYP3A4*22 and changes in ΔRUV_1–4_ or ΔQmax_1–4_, despite its association with IPSSs, suggests that genetic factors may differentially influence subjective symptoms versus objective urinary parameters. This finding highlights the complex relationship between genetic factors, drug concentration, and functional outcomes in BPH treatment.

Our study included the analysis of UGT2B7 variants—the key phase II metabolism enzyme responsible for silodosin glucuronidation. This process converts the lipophilic compound into a water-soluble metabolite (silodosin-glucuronide), facilitating renal and biliary excretion [20]. While UGT2B7 genetic variants can significantly affect enzyme activity, their role in silodosin therapy hadn’t been previously examined in clinical cohorts. None of the studied UGT2B7 variants (rs73823859 C>T, rs7439366 C>T, and rs7668282 A>G) showed significant associations with silodosin plasma concentrations (*p* > 0.05). This contrasts with findings by Wang Z et al. (2013) in healthy Chinese volunteers, where UGT2B7*1/*2 and *2/*2 genotypes showed a 27.1% and 22.7% longer terminal half-life (t_1/2_) and 37.9% and 25.2% greater AUC_0-∞_ compared to UGT2B7*1/*1, indicating a slower metabolism and increased drug exposure [17]. These discrepancies may be explained by the fact that the frequency of the T allele at rs7439366 (associated with UGT2B7*2) in our Caucasian cohort (58.3%) differs from Asian populations. Alternatively, the presence of BPH and other comorbidities in our patients may have neutralized the effect of the marker.

Our study revealed that patients homozygous for the UGT2B7 rs7439366 TT genotype (n = 34) demonstrated a significantly greater improvement in the Qmax by day 56 of treatment compared to the CC (n = 17) and CT (n = 52) genotypes: 5.4 mL/s versus 3.3 mL/s and 2.0 mL/s, respectively (*p* = 0.047). Similarly, absolute Qmax values on day 56 were higher in TT carriers: 12.75 mL/s versus 12.5 mL/s (CC) and 11.0 mL/s (CT) (*p* = 0.043). The rs73823859 C>T (MAF = 0.5%) and rs7668282 A>G (MAF = 5.3%) polymorphisms showed no significant associations with either pharmacodynamic or pharmacokinetic parameters of silodosin, which may reflect their lower functional impact or insufficient statistical power in our sample size to detect effects.

The CYP3A5*3 allele (rs776746, A6986G) causes splicing defects resulting in nonfunctional CYP3A5 protein. The CYP3A5*3 allele frequency varies substantially across populations (24–92%), explaining the observed ethnic variability in CYP3A5 expression. While poor CYP3A4 metabolizers are rare, CYP3A5 non-functionality is common in many populations [34]. Individuals carrying at least one CYP3A5*1 allele exhibit a CYP3A5 mRNA expression comparable to CYP3A4, potentially significantly influencing overall CYP3A activity [35]. In our study, GG homozygotes (poor metabolizers) showed significantly better symptom improvement (ΔIPSS_1–4_: −6.25 ± 4.60 vs. 0.0 ± 9.53 in A-allele carriers; *p* = 0.042), with sustained effects at day 28 (IPSS 7.64 ± 4.50 vs. 20.0 ± 6.55; *p* < 0.001). This aligns with CYP3A5′s role as an alternative metabolic pathway—functional CYP3A5 (in A-allele carriers) may increase clearance, reducing drug exposure and efficacy. However, we found no CYP3A5*3 effect on silodosin Css min. This contradicts previous findings by Wang Z et al. (2013), who showed that in healthy volunteers, the CYP3A5 marker significantly correlated with changes in silodosin Tmax (*p* = 0.039), with a significantly prolonged time to peak concentration (Cmax) in individuals with the CYP3A5*1/*1 genotype compared to CYP3A5*1/*3 and *3/*3 carriers [17]. Larger studies are needed to clarify CYP3A5*3′s clinical impact in BPH patients.

No significant associations were found between CYP3A4*1B and silodosin efficacy parameters, Css min concentrations, or safety outcomes. The pharmacogenetic data on CYP3A4*1B remain contradictory. While in vitro studies suggest this variant increases CYP3A activity [36], and clinical studies report its impact on tacrolimus dosing in transplant patients [37,38], its tight linkage with functional CYP3A5*1 makes it difficult to isolate CYP3A4*1B’s independent effects. The clinical relevance of CYP3A4*1B for silodosin response requires further investigation.

P-glycoprotein (P-gp), encoded by the ABCB1 gene, is a transporter protein responsible for cellular efflux of xenobiotics and pharmaceutical compounds. In vitro studies have demonstrated that silodosin is a P-gp substrate [39]. Prescribing information for silodosin contains warnings about drug interactions, noting that coadministration with strong transporter inhibitors (e.g., amiodarone, verapamil) significantly increases drug concentrations [40]. Three major ABCB1 variants—3435T>C, 2677T>G/A, and 1236T>C—have been extensively studied. These SNPs form haplotypes associated with an altered P-gp function, typically resulting in reduced activity and modified substrate specificity [41]. These variants influence pharmacokinetics and clinical outcomes for multiple drug classes including anticoagulants, antineoplastic agents, and proton pump inhibitors [42,43,44]. The rs4148738 C>T variant affects dabigatran pharmacokinetics [45], while in apixaban-treated patients, it correlates with an increased bleeding risk [46]. For silodosin, theoretically, reduced P-gp efflux activity in polymorphism carriers could increase drug concentrations in prostate/urethral tissues, potentially enhancing smooth muscle relaxation and bladder emptying. However, our study found no statistically significant effects of 3435T>C, 2677G>T, 1236C>T, or rs4148738 C>T polymorphisms on silodosin’s Css min, efficacy parameters, or safety profile in BPH patients. These findings suggest a limited clinical relevance of these genetic variants for silodosin therapy.

A key finding of our study was the absence of statistically significant associations between the investigated genetic polymorphisms (CYP3A4, CYP3A5, UGT2B7, and ABCB1) and the frequency or nature of silodosin’s ADRs. Despite significantly higher Css min plasma concentrations in CYP3A4*22 allele carriers (which theoretically could increase dose-dependent effects), no reliable association was found between this marker and systemic ADRs such as orthostatic hypotension or dizziness. Silodosin exhibits an exceptionally high selectivity (583-fold greater than tamsulosin) for α1A-adrenoceptors predominant in the prostate, bladder neck, and prostatic urethra smooth muscle, while demonstrating a minimal affinity for α1B (vascular) and α1D (CNS, myocardial) adrenoreceptors. This selectivity minimizes systemic effects even with elevated plasma drug exposure. Our data confirm this: orthostatic hypotension occurred in only four patients (26.6% of all ADRs) and dizziness in two (13.3%). The most frequent ADRs were rhinitis (33.3%) and retrograde ejaculation (13.3%)—consequences of silodosin’s local pharmacological action on α1A-adrenoreceptors in seminal ducts and nasal mucosa vasculature rather than systemic toxicity. These effects depend on the target tissue receptor blockade achieved at therapeutic doses and remain relatively insensitive to moderate increases in the systemic concentration. The lack of associations may partially reflect the insufficient power to detect effects of rare alleles (e.g., only six patients carried CYP3A4*22) on relatively uncommon or severe ADRs.

**Limitations.** This study has several important limitations that should be considered when interpreting the results. First, the relatively small sample size (n = 103) and low minor allele frequencies of key polymorphisms (CYP3A4*22 [MAF = 2.9%, n = 6 CT] and CYP3A5*1 [MAF = 3.4%, n = 7 AA/AG]) limited the statistical power to detect moderate effects and perform stratified subgroup analyses. Second, focusing solely on trough plasma concentrations (Css min) without assessing the complete pharmacokinetic profile (AUC_0-t_, Tmax, and t_1/2_) may have underestimated genetic variants’ contribution to systemic exposure. Third, the lack of strict dietary control (due to outpatient monitoring conditions) and precise documentation of concomitant medication timing (CYP3A4 substrates: amlodipine [30.1%], losartan [29.1%], and atorvastatin [18.4%]) may have introduced unaccounted variability in silodosin concentrations. Fourth, the 8-week observation period was sufficient for evaluating the symptomatic response but might be insufficient for assessing long-term outcomes (BPH progression, need for surgical intervention) and potential cumulative effects of pharmacogenetic markers on safety. Fifth, the study was conducted exclusively in Caucasian patients, limiting generalizability to other populations. In addition, polymorphisms in alcohol and aldehyde dehydrogenase genes (ADH and ALDH), though involved in silodosin metabolism, were not analyzed in this study. Previous research has shown no significant association between these variants and pharmacokinetic parameters [17], but their contribution cannot be completely excluded.

## 5. Conclusions

This study demonstrates that genetic polymorphisms CYP3A4*22 (rs35599367) and CYP3A5*3 (rs776746) are associated with significant differences in the Css min of silodosin in BPH patients, which may reflect altered drug exposure, although a full pharmacokinetic profiling (e.g., Cmax, Tmax, and Cl) was not conducted. Another finding that these markers have clinical effects of silodosin in BPH patients is as follows: carriers of the CYP3A4*22 allele (CT genotype) had higher Css min values compared to non-carriers, which correlate with less improvement in absolute IPSSs at all visits (*p* < 0.05) but not with symptom dynamics (ΔIPSS_1–4_, *p* = 0.939), likely due to more severe symptoms initially in this subgroup; meanwhile, CYP3A5*3 GG homozygotes showed a significantly better IPSS improvement by day 56 (ΔIPSS_1–4_ −6.25 ± 4.60 versus 0.0 ± 9.53 in AA/AG; *p* = 0.042) and at day 28 (IPSS 7.64 ± 4.50 versus 20.0 ± 6.55; *p* < 0.001), indicating CYP3A5′s role in silodosin metabolism. The UGT2B7 rs7439366 TT variant was associated with better Qmax dynamics (ΔQmax_1–4_ 5.4 mL/s versus 3.3 in CC and 2.0 in CT; *p* = 0.041), highlighting glucuronidation’s contribution to urodynamic responses. ABCB1 polymorphisms showed no significant associations with drug efficacy or safety in BPH patients. Despite the identified differences, no marker was associated with ADR frequency (15 events in 42 patients; *p* > 0.05), attributable to silodosin’s high uroselectivity minimizing systemic effects even at an increased exposure.

The studied polymorphisms showed limited clinical significance with moderate effects. In our study, no statistically significant associations were found between genetic variants and ADRs. However, the relatively small sample size, low number of ADRs, limited observation period, and imprecise documentation of concomitant therapies significantly limit the power to detect such associations. Therefore, while our data do not demonstrate a link between pharmacogenetic markers and ADRs, these findings should be interpreted with caution. Larger prospective studies with longer follow-up and sufficient ADR event rates are warranted to more accurately assess the clinical relevance of pharmacogenetic testing in this context.

## Figures and Tables

**Table 1 jpm-15-00386-t001:** Clinical and demographic characteristics of patients included in the study.

Parameters		Overall Patient Cohort (n = 103)
Age, years		69.5 [64;76]
BMI, kg/m^2^		28.72 [26.99;31.49]
Smoking (n)		11
Alcohol (n)		7
Laboratory Parameters at Baseline		
Creatinine, mmol/L		91.0 [81.0;103.0]
Urea, mmol/L		5.95 [4.8;6.9]
Relative density, g/L		1019.5 [1015.0;1025.0]
pH		6.0 [5.5;6.0]
Hemoglobin, g/L		148.5 [145.0;155.0]
Red blood cells, 10^9^/L		4.83 [4.64;5.2]
White blood cells, 10^9^/L		6.27 [5.56;6.99]
Platelets, 10^9^/L		222.0 [180.0;262.0]
Erythrocyte sedimentation rate, mm/hour		5.0 [4.0;9.0]
Prostate-specific antigen, ng/ml		1.89 [0.86;3.36]
Comorbidities, n (%)		
Cardiovascular:		65 (63.11%)
Hypertension		40 (38.83%)
Ischemic heart disease		20 (19.42%)
Others		5 (4.85%)
Endocrine:		9 (8.74%)
Type 2 diabetes mellitus—insulin-independent		7 (6.80%)
Hypothyroidis		2 (1.9%)
Pulmonary:		2 (1.94%)
Chronic obstructive pulmonary disease		1 (0.97%)
Bronchial asthma		1 (0.97%)
Gastroenterological:		3 (2.9%)
Gastroesophageal reflux disease		1 (0.97%)
Hepatic steatosis		2 (1.94%)
Urological:		10 (9.71%)
Urolithiasis		3 (2.91%)
Renal cyst		5 (4.85%)
Erectile dysfunction		2 (1.94%)
Neurological:		7 (6.8%)
Degenerative-dystrophic spinal diseases		3 (2.91%)
Intervertebral hernias		4 (3.88%)
Concomitant Medication Therapy, n (%)		
*Category*	*Medication*	*n (%)*
CYP3A4 inhibitors	Allapinine	1 (0.97%)
	Amiodarone	2 (1.94%)
	Eplerenone	1 (0.97%)
CYP3A4 inducers	-	-
CYP3A4 substrates	Amlodipine	31 (30.1%)
	Atorvastatin	19 (18.45%)
	Losartan	30 (29.13%)
	Nifedipine	9 (8.74%)
	Doxylamine	1 (0.97%)

Note: Continuous variables are presented as median [IQR] unless otherwise indicated; categorical variables are presented as number (percentage).

**Table 2 jpm-15-00386-t002:** Genotype distribution of studied polymorphisms: allele frequencies and Hardy–Weinberg equilibrium testing.

Allele Variants	Genotypes	n (%)	Minor Allele Frequency, %	χ^2^	*p*-Value
CYP3A4*1B (rs2740574, A>G)	AA	99 (96.12)	1.9	0.040	0.980
	AG	4 (3.88)			
	GG	0 (0)			
CYP3A4*22 (rs35599367, C>T)	CC	97 (94.17%)	2.9	0.093	0.955
	CT	6 (5.83%)			
	TT	0 (0)			
CYP3A5*3 (rs776746, A6986G)	AA	1 (0.97%)	96.1	4.971	0.083
	AG	6 (5.83%)			
	GG	96 (93.20%)			
UGT2B7 (rs73823859, C>T)	CC	102 (99.03)	0.5	0.002	0.999
	CT	1 (0.97)			
	TT	0 (0)			
UGT2B7 (rs7439366, C>T)	CC	17 (16.50%)	58.3	0.149	0.928
	CT	52 (50.49%)			
	TT	34 (33.01%)			
UGT2B7 (rs7668282, A>G)	AA	92 (89.32%)	5.3	0.328	0.849
	AG	11 (10.68%)			
	GG	0 (0)			
ABCB1 (rs4148738, C>T)	CC	28 (27.18)	44.2	2.683	0.261
	CT	59 (57.28)			
	TT	16 (15.54)			
ABCB1 (rs1045642, 3435T>C)	TT	45 (43.69%)	35.4	0.791	0.673
	CT	43 (41.75%)			
	CC	15 (14.56%)			
ABCB1 (rs2032582, 2677G>T)	GG	19 (18.45%)	55.3	0.379	0.827
	GT	54 (52.43%)			
	TT	30 (29.12%)			
ABCB1 (rs1128503, 1236C>T)	CC	13 (12.62)	55.8	8.047	0.018
	CT	65 (63.11)			
	TT	25 (24.27)			

**Table 3 jpm-15-00386-t003:** Effects of genetic polymorphisms on silodosin Css min and IPSSs at baseline, day 14, day 28, and day 56 of treatment.

Marker	Genotypes	n	Css min, ng/mL	Visits								ΔIPSS_1–4_	ΔQoL_1–4_
				Visit 1 (Day 1)		Visit 2 (Day 14)		Visit 3 (Day 28)		Visit 4 (Day 56)			
				IPSS, Points	QoL, Points	IPSS, Points	QoL, Points	IPSS, Points	QoL, Points	IPSS, Points	QoL, Points		
CYP3A4*1B (rs2740574, A>G)	AA	99	9.45	13.85 ± 5.48	4.29 ± 0.64	9.58 ± 4.83	2.92 ± 1.25	8.63 ± 5.61	2.31 ± 1.27	7.31 ± 5.93	1.95 ± 1.49	−5.73 ± 5.21	−2.19 ± 1.43
	AG	4	10.62	12 ± 2.45	4 ± 2.59	5 ± 2.29	3 ± 2.60	4 ± 2.80	3 ± 1.05	3 ± 3.44	2 ± 1.55	−9 ± 4.22	−2 ± 0.61
	*p*-value		0.914	0.347	0.185	0.232	0.911	0.817	0.358	0.982	0.898	0.289	0.821
CYP3A4*22 (rs35599367, C>T)	CC	97	9	14.21 ± 4.94	4.21 ± 0.52	9.21 ± 4.60	2.78 ± 1.11	7.94 ± 5.38	2.18 ± 1.15	6.55 ± 5.66	1.81 ± 1.39	−5.78 ± 5.29	−2.23 ± 1.44
	CT	6	17.59	18.5 ± 7.18	5.0 ± 1.15	15.0 ± 5.09	4.25 ± 1.70	13.0 ± 4.14	3.5 ± 1.73	11.5 ± 4.65	3.25 ± 1.89	−6.0 ± 4.54	−1.75 ± 1.25
	*p*-value		**0.049 ***	0.065	0.162	**0.047 ***	**0.022 ***	**0.037 ***	**0.045 ***	**0.023 ***	**0.045 ***	0.939	0.520
CYP3A5*3 (rs776746, A6986G)	AA + AG ^$^	7	8.4	21.0 ± 7.21	4.66 ± 1.15	12.67 ± 11.15	2.67 ± 2.51	20.0 ± 6.55	3.33 ± 1.15	11.67 ± 10.21	2.0 ± 1.73	0.0 ± 9.53	−1.0 ± 2.0
	GG	96	9.58	13.25 ± 4.96	4.25 ± 0.59	9.23 ± 4.20	2.94 ± 1.14	7.64 ± 4.50	2.23 ± 1.24	6.87 ± 5.51	1.94 ± 1.48	−6.25 ± 4.60	−2.28 ± 1.35
	*p*-value		0.770	0.015	0.286	0.239	0.708	**0.000066 ***	0.145	0.178	0.954	**0.042 ***	0.133
UGT2B7 (rs73823859, C>T)	CC	102	9.55	13.80 ± 5.42	4.28 ± 0.63	9.47 ± 4.82	2.92 ± 1.23	8.52 ± 5.59	2.30 ± 1.25	7.21 ± 5.90	1.95 ± 1.48	−5.80 ± 5.17	−2.19 ± 1.41
	CT	1	7.00	17	4	9	3	10	3	9	3	−6	−1
	*p*-value		0.610	0.370	0.249	0.280	0.889	0.223	0.154	0.803	0.708	0.354	0.346957
UGT2B7 (rs7439366, C>T)	CC	17	10.44	15.0 ± 6.50	4.14 ± 0.37	11.28 ± 5.12	3.14 ± 0.69	8.85 ± 5.08	2.14 ± 1.06	8.42 ± 6.24	2.42 ± 0.97	−6.57 ± 4.50	−1.71 ± 1.11
	CT	52	8.49	13.9 ± 5.55	4.38 ± 0.74	8.71 ± 4.96	2.80 ± 1.53	8.95 ± 6.07	2.61 ± 1.24	7.19 ± 6.05	2.04 ± 1.74	−5.14 ± 6.15	−2.04 ± 1.46
	TT	34	10.57	13.07 ± 4.93	4.21 ± 0.57	9.71 ± 4.54	3.0 ± 0.96	7.71 ± 5.38	1.92 ± 1.32	6.64 ± 5.86	1.57 ± 1.22	−6.42 ± 3.93	−2.64 ± 1.44
	*p*-value		0.613	0.749	0.617	0.473	0.806	0.809	0.268	0.815	0.429	0.714	0.304
UGT2B7 (rs7668282, A>G)	AA	92	9.56	13.71 ± 5.01	4.23 ± 0.58	9.55 ± 4.40	2.94 ± 1.16	8.44 ± 5.58	2.26 ± 1.26	7.23 ± 5.74	1.94 ± 1.50	−5.73 ± 5.40	−2.15 ± 1.48
	AG	11	8.98	14.75 ± 9.53	4.75 ± 0.95	8.75 ± 8.84	2.75 ± 2.06	9.25 ± 6.5	2.75 ± 1.25	7.0 ± 8.28	2.0 ± 1.41	−6.5 ± 2.38	−2.5 ± 0.57
	*p*-value		0.869	0.720	0.126	0.755	0.765	0.788	0.468	0.940	0.947	0.782	0.651
ABCB1 (rs4148738, C>T)	CC	28	9.16	12.91 ± 3.26	4.25 ± 0.62	9.08 ± 3.34	3.08 ± 1.08	6.58 ± 4.07	2.0 ± 1.27	6.41 ± 5.9	1.83 ± 1.69	−6.5 ± 5.28	−2.41 ± 1.37
	CT	59	9.53	13.79 ± 5.92	4.33 ± 0.7	10.91 ± 5.58	3.04 ± 1.16	9.58 ± 6.22	2.33 ± 1.27	9.16 ± 7.07	2.20 ± 1.47	−4.62 ± 4.55	−2.12 ± 1.42
	TT	16	9.96	15.66 ± 7.08	4.16 ± 0.40	10.0 ± 2.82	3.5 ± 0.54	8.16 ± 5.19	2.83 ± 1.16	6.5 ± 5.0	2.16 ± 1.32	−9.16 ± 6.43	−2.0 ± 1.67
	*p*-value		0.974	0.609	0.832	0.550	0.645	0.319	0.422	0.416	0.779	0.135	0.800
ABCB1 (rs1045642, 3435T>C)	CC	45	10.6	13.0 ± 7.77	4.16 ± 0.4	7.83 ± 4.07	2.66 ± 1.03	7.16 ± 4.07	2.83 ± 1.47	5.83 ± 4.26	1.83 ± 1.60	−7.16 ± 6.91	−2.33 ± 1.5
	TC	43	9.35	14.64 ± 6.4	4.35 ± 0.7	11.58 ± 5.18	3.11 ± 1.05	9.52 ± 5.53	2.35 ± 1.05	8.7 ± 5.94	2.11 ± 1.05	−5.94 ± 3.86	−2.23 ± 1.39
	TT	15	6.62	13.31 ± 3.55	4.26 ± 0.65	9.84 ± 4.29	3.26 ± 1.09	8.05 ± 6.13	2.1 ± 1.37	8.05 ± 7.61	2.15 ± 1.83	−5.26 ± 5.76	−2.1 ± 1.48
	*p*-value		0.444	0.609	0.832	0.550	0.645	0.319	0.422	0.416	0.779	0.737	0.932
ABCB1 (rs2032582, 2677G>T)	GG	19	9.03	15.0 ± 6.70	4.14 ± 0.37	9.85 ± 2.60	3.42 ± 0.53	7.85 ± 4.81	2.71 ± 1.11	6.28 ± 4.60	2.14 ± 1.21	−8.71 ± 5.99	−2.0 ± 1.52
	GT	54	9.97	14.0 ± 6.15	4.36 ± 0.72	11.04 ± 5.82	3.09 ± 1.19	9.72 ± 6.46	2.27 ± 1.27	9.45 ± 7.32	2.31 ± 1.46	−4.54 ± 4.75	−2.04 ± 1.43
	TT	30	8.92	12.84 ± 3.13	4.23 ± 0.59	9.15 ± 3.21	3.0 ± 1.08	6.84 ± 4.01	2.15 ± 1.34	6.46 ± 5.65	1.69 ± 1.70	−6.38 ± 5.07	−2.53 ± 1.39
	*p*-value		0.895	0.689	0.687	0.512	0.690	0.326	0.635	0.322	0.497	0.160	0.577
ABCB1 (rs1128503, 1236C>T)	CC	13	9.75	15.0 ± 6.70	4.14 ± 0.37	9.85 ± 2.60	3.42 ± 0.53	7.85 ± 4.81	2.71 ± 1.11	6.28 ± 4.60	2.14 ± 1.21	−9.0 ± 7.17	−1.4 ± 0.89
	TC	65	9.36	13.61 ± 5.72	4.34 ± 0.68	10.76 ± 5.38	3.03 ± 1.11	9.19 ± 6.13	2.26 ± 1.22	8.73 ± 6.97	2.07 ± 1.49	−4.88 ± 4.5	−2.26 ± 1.48
	TT	25	9.71	13.09 ± 3.36	4.27 ± 0.64	9.09 ± 3.50	3.09 ± 1.13	6.72 ± 4.24	2.0 ± 1.34	6.54 ± 6.17	1.9 ± 1.75	−6.54 ± 5.53	−2.36 ± 1.43
	*p*-value		0.993	0.516	0.546	0.622	0.566	0.473	0.205	0.642	0.699	0.232	0.417

Note: ^$^—AA and AG genotypes were grouped due to the low frequency of AA and a similar functional classification as CYP3A5 expressors. Abbreviations: IPSS—International Prostate Symptom Score, QoL—quality of life, and Css min—trough steady-state plasma concentration; ∆IPSS_1–4_ represents changes in IPSSs relative to baseline until day 56; ∆QoL_1–4_ indicates changes in quality of life subscale (QoL) scores relative to baseline until day 56; * statistically significant *p*-values were observed, *p*-value <0.05 (Bold); and data are presented as mean ± standard deviation (SD) unless otherwise indicated.

**Table 4 jpm-15-00386-t004:** Impact of studied polymorphisms on urodynamic parameters (prostate volume, residual urine volume, and Qmax).

Marker	Genotype	n	Visits						ΔRUV_1–4_	ΔQmax_1–4_
			Visit 1 (Day 1)			Visit 4 (Day 56)				
			PV, cm^3^	RUV, ml	Qmax, ml/sec	PV, cm^3^	RUV, mL	Qmax, ml/sec		
CYP3A4*1B (rs2740574, A>G)	AA	99	37.2 [31.1;45.3]	14.0 [6.0;33.0]	9.3 [8.1;11.4]	36.5 [31.2;46.0]	13.0 [3.0;25.0]	11.2 [9.2;13.5]	0.0 [−10.0;13.0]	2.4 [0.8;4.8]
	AG	4	45.2 [37.65;53.5]	32.5 [15.0;5.0]	9.05 [7.7;10.0]	39.7 [28.6;54.3]	20.0 [7.5;51.5]	11.25 [9.25;15.9]	2.5 [−6.5;12.5]	2.2 [1.55;5.9]
	*p*-value		0.804	0.398	0.695	0.858	0.657	0.289	0.706	0.865
CYP3A4*22 (rs35599367, C>T)	CC	97	36.6 [31.1;47.2]	12.0 [31.2;46.0]	9.35 [7.8;11.4]	36.25 [31.2;46.0]	12.75 [6.0;30.0]	12.1 [9.1;14.0]	0.0 [−8.0;13.0]	2.55 [0.8;4.8]
	CT	6	38.75 [32.3;50.45]	27.0 [7.0;52.5]	8.15 [8.1;10.0]	39.5 [32.35;50.5]	20.0 [5.0;54.0]	10.1 [9.15;10.65]	0.0 [−11.5;12.0]	2.0 [0.8;2.5]
	*p*-value		0.863	0.453	0.622	0.592	0.715	0.293	0.830	0.492
CYP3A5*3 (rs776746, A6986G)	AA + AG ^$^	7	75.9 [26.3;95.3]	30.0 [6.0;40.0]	9.9 [8.1;14.3]	89.7 [26.3;105.0]	10.0 [0.0;30.0]	10.8 [10.1;14.0]	10.0 [6.0;20.0]	0.9 [0.3;2.0]
	GG	96	37.2 [31.1;45.3]	13.0 [5.0;33.0]	9.2 [7.8;11.4]	36.5 [31.2;45.0]	13.3 [3.0;25.0]	11.2 [8.8;13.5]	0.0 [−10.0; −13.0]	2.7 [0.8;5.9]
	*p*-value		0.305	0.557	0.574	0.241	0.660	0.769	0.187	0.222
UGT2B7 (rs73823859, C>T)	CC	102	37.25 [31.1;47.2]	13.5 [6.0;33.0]	9.25 [8.0;11.4]	36.75 [31.2;46.0]	12.75 [6.0;33.0]	11.0 [9.1;13.5]	0.0 [−10.0;13.0]	2.2 [0.8;4.8]
	CT	1	37	14	9	36	14	10	0.0	1.0
	*p*-value		0.847	0.363	0.592	0.999	0.702	0.289	0.962	0.878
UGT2B7 (rs7439366, 802 C>T)	CC	17	31.1 [21.4;39.2]	14.0 [0.0;20.0]	9.3 [9.0;15.8]	30.0 [27.0;40.6]	15.0 [0.0;30.0]	12.5 [10.1; 27.6]	0.0 [−10.0;13.0]	3.3 [0.8;11.0]
	CT	52	37.3 [33.4;45.3]	7.5 [0.0;35.0]	9.4 [7.6;11.4]	37.3 [33.0;44.0]	11.0 [0.0;33.0]	11.0 [0.0;30.0]	0.0 [−10.0;10.0]	2.0 [0.7;3.0]
	TT	34	44.05 [34.0;61.7]	22.5 [10.0;35.0]	6.5 [4.0;12.0]	41.0 [32.0;49.2]	12.75 [10.0;25.0]	12.75 [10.1;14.0]	6.5 [−8.0;20.0]	5.4 [0.9;9.6]
	*p*-value		0.089	0.638	0.695	0.213	0.679	**0.043 ***	0.527	**0.041 ***
UGT2B7 (rs7668282, A>G)	AA	92	37.8 [31.5;47.2]	14.5 [6.0;33.0]	9.35 [7.8;11.4]	36.75 [31.2;46.0]	14.0 [3.0;25.0]	10.5 [8.8;13.5]	0.0 [−10.0;14.0]	2.0 [0.7;4.8]
	AG	11	33.85 [28.4;43.35]	3.0 [0.0;23.0]	8.9 [8.35;9.5]	33.85 [28.0;42.65]	7.5 [2.5;20.0]	7.5 [2.5;20.0]	0.5 [−5.0;5.5]	3.15 [2.5;6.8]
	*p*-value		0.506	0.238	0.699	0.592	0.440	0.324	0.897	0.238
ABCB1 (rs4148738, C>T)	CC	28	42.4 [32.6;48.35]	15.0 [6.75;32.5]	9.0 [8.0;11.45]	37.3 [31.2;45.0]	11.75 [7.5;27.5]	10.95 [8.6;15.65]	0.5 [−7.15;7.5]	1.3 [0.6;7.2]
	CT	59	37.25 [32.15;46.15]	12.5 [6.0;31.5]	9.4 [8.05;11.5]	36.25 [29.5;46.7]	11.5 [1.0;23.5]	11.0 [9.7;13.75]	5.5 [−9.0;17.0]	2.85 [0.85;5.35]
	TT	16	33.35 [30.5;45.3]	6.5 [0.0;64.0]	8.65 [7.4;10.0]	35.05 [29.7;48.2]	42.5 [0.0;78.5]	11.25 [8.7;12.5]	1.0 [−13.0;0.0]	2.2 [1.3;3.33]
	*p*-value		0.808	0.420	0.623	0.863	0.057	0.651	0.369	0.310
ABCB1 (rs1045642, 3435T>C)	CC	15	33.85 [27.6;45.1]	0.0 [0.0;10.0]	9.9 [9.2;10.0]	27.85 [25.4;38.0]	6.0 [0.0;25.0]	12.65 [12.4;13.5]	0.0 [−10.0;5.0]	2.9 [2.4;3.3]
	TC	43	38.3 [34.0;61.7]	30.0 [9.0;40.0]	8.6 [8.1;9.9]	37.5 [34.0;53.1]	15.0 [10.0;35.0]	10.2 [10.1;13.4]	0.0 [−5.0;20.0]	2.0 [0.9;5.9]
	TT	45	35.2 [29.1;47.2]	11.0 [6.0;30.0]	9.4 [7.7;11.8]	36.5 [31.2;44.0]	11.0 [5.0;25.0]	10.1 [8.4;14.0]	1.0 [−10.0;11.0]	1.5 [0.3;4.8]
	*p*-value		0.168	0.060	0.548	0.116	0.813	0.950	0.253	0.764
ABCB1 (rs2032582, 2677G>T)	GG	19	35.2 [30.5;61.7]	13.0 [0.0;65.0]	8.6 [7.4;10.0]	37.5 [29.7;48.4]	15.0 [0.0;78.0]	12.4 [8.7;12.7]	0.0 [−13.0;22.0]	2.4 [1.3;4.1]
	GT	54	36.6 [31.1;44.3]	10.5 [6.0;30.0]	9.4 [8.0;11.4]	36.25 [32.0;45.0]	10.0 [0.0;22.0]	10.5 [9.5;13.5]	3.0 [−10.0;14.0]	2.75 [0.8;4.8]
	TT	30	43.8 [35.0;47.2]	15.0 [7.5;30.0]	9.4 [8.3;11.4]	37.0 [31.2;44.0]	12.5 [10.0;25.0]	11.8 [8.8;18.9]	1.0 [−4.3;5.0]	1.5 [0.7;9.4]
	*p*-value		0.969	0.292	0.561	0.602	0.114	0.834	0.616	0.876
ABCB1 (rs1128503, 1236C>T)	CC	13	31.5 [30.5;45.3]	15.0 [6.0;30.0]	9.2 [8.1;10.0]	32.6 [29.7;48.2]	12.0 [5.0;30.0]	12.4 [10.1;12.5]	−0.5 [−7.15;7.5]	2.4 [2.0;3.3]
	TC	65	36.6 [31.1;44.3]	10.5 [6.0;30.0]	9.4 [8.0;11.4]	36.25 [32.0;45.0]	10.0 [0.0;22.0]	10.5 [9.5;13.5]	3.0 [−10.0;14.0]	2.75 [0.8;4.8]
	TT	25	43.8 [35.0;49.5]	15.0 [0.0;13.0]	9.4 [8.3;11.5]	37.6 [31.2;46.0]	9.0 [0.0;30.0]	11.8 [8.8;18.9]	1.0 [−10.0;5.0]	1.5 [0.5;10.3]
	*p*-value		0.860	0.704	0.794	0.825	0.405	0.794	0.053	0.869

Note: ^$^—AA and AG genotypes were grouped due to the low frequency of AA and a similar functional classification as CYP3A5 expressors. Abbreviations: PV—prostate volume; RUV—residual urine volume; Qmax—maximum urinary flow rate; ΔRV_1–4_—change in RUV from baseline to day 56; ΔQmax_1–4_—change in Qmax from baseline to day 56; * statistically significant *p*-values were observed, *p*-value < 0.05 (Bold); data are presented as median [IQR] unless otherwise indicated; and delta (Δ) values were calculated based on individual patient-level differences between Visit 4 and Visit 1 and are summarized as median [IQR].

## Data Availability

The datasets generated and analyzed during this study are not publicly available due to ethical restrictions and patient confidentiality protections under Russian Federation laws on personal data protection (Federal Law No. 152-FZ). However, anonymized data supporting the findings may be made available upon reasonable request from qualified researchers, subject to approval by the Local Ethics Committee of the Russian Medical Academy of Continuous Professional Education (contact: rmapo@rmapo.ru). Requests should include a detailed research proposal and data protection plan.

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
