# Peer review of "Genetic Modulation of Silodosin Exposure and Efficacy: The Role of CYP3A4, CYP3A5, and UGT2B7 Polymorphisms in Benign Prostatic Hyperplasia Management"

_jpm, 2025, doi:10.3390/jpm15080386_

Round 1

Reviewer 1 Report

Comments and Suggestions for Authors

The study by Abdullaev and colleagues was a prospective observational trial of 103 Russian men treated with silodosin in the setting of benign prostatic hyperplasia.  Genotyping of relevant polymorphisms of CYP3A4, CYP3A5, UGT2B7 and ABCB1 was completed, and drug concentrations and urodynamic data were collected.  The study had notable findings for CYP3A4*22, CYP3A5*3, and UGT2B7 rs7439366C>T polymorphisms. 

This was a well-written manuscript and a carefully conducted study. The study had proper inclusion/exclusion criteria, and the authors selected appropriate genetic polymorphisms associated with silodosin metabolism.  Analytic and statistical methods seem appropriate.

There were several discrepancies noted between the data tables and results section, and this will need to be updated prior to publication.  Additionally, this reviewer had a concern about one of the conclusions from the study, specifically when applying the adverse event data to clinical practice. 

I also noted a couple of sentences that seemed to similar to the reference sources.  I'm asking that those sentences be revised out of an abundance of caution.  

It was a pleasure reading your work, and I hope the following changes are helpful in improving your manuscript.

Abstract

Line 31 - CYP3A5*3 is missing the asterisk.

Introduction

Line 48-50 – This sentence (“Global incidence and prevalence rates are projected…”) is very similar to the reference source and might be considered plagiarism by some standards.  I would recommend revising the sentence, summarizing the data a little more. 

Line 50-51 – The reference supporting this sentence (ref #5) does contain data addressed in this sentence.  The reference discusses data from the UK, the text in your manuscript discusses Medicare (US) data.  Please update with an appropriate reference.  Also, the data makes a leap from Medicare expenditures to worldwide costs.  I cannot see the reference, but I infer that the study extrapolated the US data.  The sentence needs the word “extrapolated” added somewhere for clarity. 

Line 59-62 – These data are reported in a very similar way (almost verbatim) to the source reference.  This sentence should be revised and the data summarized to avoid any concerns. 

Line 87 to line 110 – This paragraph appears to be duplicated.  Please remove.

Materials and Methods

Line 181 – This paragraph discusses steady-state concentration determination and specifies samples were collected no earlier than 5 days after drug initiation.  However, it is not clear when the dose was collected relative to the dose.  Given the half life is 11-13 hours, this could be a significant confounder in Css.  Please clarify when samples were collected relative to the last dose.  If unknown or unspecified, this should be stated. 

Results

Line 211-213 – The percentages for comorbidities in this sentence do not match the numbers in Table 1.  Please clarify and/or update the text as appropriate.

Line 257-261 – This sentence states the data is for the 3435T>C polymorphism, however, the data maps to rs4148738 in Table 3.  Please clarify and update. 

Line 267 and line 363 – The p-value (p=0.044) listed is different than data in Table 3 (p=0.043).  This same discrepancy is found in both the results (line 267) and discussion (line 363).  Please clarify and update. 

Line 278 – Table 4 lists delta RUV for the CC diplotype as -0.5.  The text lists +0.5.  Please clarify and update.

Line 288 – The p-values do not match the data in Table 4 for delta RUV and delta Qmax.  Please clarify and update.

Discussion

Line 324 – CYP3A4*22 is missing the asterisk.

Conclusions

Line 459-460 – This line states “No link was found between genetic variants and adverse reactions, questioning the need for routine pharmacogenetic testing…”  This is a major point of contention.  Your limitations state that the study had a relatively small sample size, imprecise documentation of concomitant drug administration, and a relatively short window of observation.  I would add that there was a relatively low number of adverse reactions found in the study population, certainly far too few to accurately assess AEs between groups. In sum, there is a high risk for a type II error here. While I would agree that no association was found in the data, the limitations of your study do not support a conclusion calling into conclusion the use of routine PGx testing on the basis of AEs.  At the very least, larger and longer studies that focus on the polymorphisms you identified are warranted.  I suggest revising this final paragraph accordingly. 

Editorial comments:

Page numbers are not sequential throughout the document. 

Tables and Figures

Table 1 – The categories and medications under Concomitant Medication Therapy are difficult to follow and may lead to some misinterpretation of data.  Please format the table so that the categories/medications align more clearly.

Table 3 and 4 – Table header or footnotes should indicate what type of data is represented.  I believe data in table 3 is mean +/- STD; table 4 is median [IQR]

Table 4 – There appears to be some discrepancy in the delta RUV and delta Qmax columns.  (1) The difference values don’t seem to calculate correctly from visit one to visit 4 for several of the data points.   For instance, in the top row, the values for Qmax are 9.3 for visit 1 and 11.2 for visit 2.  The difference is 1.9 which does not match the delta Qmax column which is 2.4.  Please clarify this discrepancy.  (2) There seems to be some disagreement between the text and the Table 4 values, specifically related to positive or negative signs in the delta columns.  For instance, line 267-268 lists delta Qmax data with negative integers, but the table has no sign listed (inferring positive).  The interquartile ranges also don’t agree with the median values in many cases… presumably the median should fall between 1st and 3rd quartile but several lines of the delta RUV and delta Qmax columns do not. 

Author Response

We would like to express our sincere gratitude to Reviewer for the careful, thorough, and constructive review of our manuscript. We highly appreciate the detailed comments and suggestions, which significantly contributed to improving the quality, clarity, and precision of our work. Each point raised was addressed with great attention, and appropriate revisions were made throughout the manuscript. Below, we provide our detailed responses to each comment, along with explanations of the corresponding changes

ABSTRACT

Line 31 - CYP3A5*3 is missing the asterisk.

Response: The asterisk has been added

INTRODUCTION

Line 48-50 – This sentence (“Global incidence and prevalence rates are projected…”) is very similar to the reference source and might be considered plagiarism by some standards.  I would recommend revising the sentence, summarizing the data a little more. 

Response: We appreciate the reviewer’s observation regarding the similarity of the sentence on lines 48-50 to the referenced source. To address this, we have revised the sentence to summarize the data in our own words, ensuring clarity and originality while preserving the scientific accuracy of the projections. The revised sentence now reads: "Projections indicate a global increase in benign prostatic hyperplasia (BPH) incidence and prevalence, rising from approximately 962 and 7,879 per 100,000 population in 2022 to about 999 and 8,621 per 100,000 by 2035 [4]." We believe this revision avoids any concerns about plagiarism while maintaining the intended meaning.

Line 50-51 – The reference supporting this sentence (ref #5) does contain data addressed in this sentence.  The reference discusses data from the UK, the text in your manuscript discusses Medicare (US) data.  Please update with an appropriate reference.  Also, the data makes a leap from Medicare expenditures to worldwide costs.  I cannot see the reference, but I infer that the study extrapolated the US data.  The sentence needs the word “extrapolated” added somewhere for clarity. 

Response: The sentence on lines 50-51 has been revised to: “Based on 2019 US Medicare data, global healthcare costs for benign prostatic hyperplasia (BPH) are estimated to be approximately $73.8 billion annually, as extrapolated from US expenditure trends [5].” This revision incorporates the term “extrapolated” to clarify that the global cost estimate is derived from US Medicare data. Additionally, we have updated the reference [5] to Launer et al. (2021), which accurately supports the global cost estimate, correcting the previous incorrect citation of Devlin et al. (2021), which focused on UK data. We believe these changes address the reviewer’s concerns and enhance the manuscript’s accuracy

Line 59-62 – These data are reported in a very similar way (almost verbatim) to the source reference.  This sentence should be revised and the data summarized to avoid any concerns. 

Response: We thank the reviewer for noting the similarity of the sentence on lines 59-62 to the source reference. To address this, we have revised the sentence to summarize the data in our own words, ensuring originality while preserving the scientific accuracy of the prevalence and temporal trends of LUTS associated with BPH. The revised sentence now reads: “A systematic review of 222 studies across 36 countries reported that approximately 63% of men experience LUTS associated with BPH, with about 31% having moderate-to-severe symptoms (95% CI: 58.0–68.1 and 28.8–33.8, respectively). The prevalence of LUTS has risen over time, from roughly 27.4% in the 1990s to 31.9% in the 2000s, and 36.2% in the 2010s (95% CI: 24.5–30.3, 27.3–36.7, and 30.7–41.9, respectively)[6].” We believe this revision mitigates concerns about verbatim similarity and maintains clarity and fidelity to the source data.

Line 87 to line 110 – This paragraph appears to be duplicated.  Please remove.

Response: We appreciate the reviewer’s identification of the duplicated paragraph on lines 87–110. We have removed this redundant section, as it was identical to the paragraph on lines 64–86.

MATERIALS AND METHODS

Line 181 – This paragraph discusses steady-state concentration determination and specifies samples were collected no earlier than 5 days after drug initiation.  However, it is not clear when the dose was collected relative to the dose.  Given the half life is 11-13 hours, this could be a significant confounder in Css.  Please clarify when samples were collected relative to the last dose.  If unknown or unspecified, this should be stated. 

Response: Clarify the timing of plasma sample collection relative to the last dose of silodosin, given its half-life of 11–13 hours. To address this, we have revised the sentence in the Materials and Methods section (line 181) to: “For all patients, the minimum steady-state plasma concentration (Css min) of silodosin was determined. Plasma samples were collected in the morning, immediately before the administration of the next daily dose, no earlier than 5 days after the initiation of silodosin treatment to ensure steady-state conditions.” This revision specifies that samples were collected just prior to the next dose, ensuring that Css min reflects trough concentrations and minimizing variability due to dosing timing. We believe this addition addresses the potential confounding factor.

Results

Line 211-213 – The percentages for comorbidities in this sentence do not match the numbers in Table 1.  Please clarify and/or update the text as appropriate.

Response: We appreciate the reviewer’s observation regarding the discrepancy between the percentages reported on lines 211–213 and those in Table 1. The original sentence inadvertently reported percentages relative to the subgroup of patients with comorbidities, which caused confusion with Table 1, where percentages are calculated relative to the total cohort (n=103). To address this, we have revised the sentence to: “The majority of patients had comorbidities, with cardiovascular conditions present in 63.1% (n=65) of the cohort, including hypertension (38.83%, n=40) and ischemic heart disease (19.42%, n=20). Additionally, type 2 diabetes mellitus was observed in 6.8% (n=7), and urological conditions in 9.71% (n=10) of patients”. This revision aligns the percentages with Table 1, ensuring consistency and clarity. We have also included absolute numbers to enhance transparency. We apologize for the oversight and believe this correction resolves the issue.

Line 257-261 – This sentence states the data is for the 3435T>C polymorphism, however, the data maps to rs4148738 in Table 3.  Please clarify and update. 

Response: We appreciate the reviewer’s identification of the discrepancy in the description of the polymorphism on lines 257–261. The original sentence incorrectly referred to the polymorphism as 3435T>C and reported an incorrect number of CC homozygotes (n=6). We have revised the sentence to: “However, for the rs4148738 C>T polymorphism, a trend toward greater IPSS improvement was noted in TT homozygotes (n=16, ΔIPSS1-4 = -9.16±6.43) compared to CT heterozygotes (n=59, ΔIPSS1-4 = -4.62±4.55) and CC homozygotes (n=28, ΔIPSS1-4 = -6.5±5.28), though these differences did not reach statistical significance (p=0.135).” This revision corrects the polymorphism identifier to rs4148738 C>T and updates the number of CC homozygotes to 28, ensuring consistency with Table 3. We apologize for the oversight and believe this correction enhances the manuscript’s accuracy

Line 267 and line 363 – The p-value (p=0.044) listed is different than data in Table 3 (p=0.043).  This same discrepancy is found in both the results (line 267) and discussion (line 363).  Please clarify and update. 

Response: We thank the reviewer for identifying the discrepancy in the p-value reported on lines 267 and 363 (p=0.044) compared to Table 3 (p=0.043). We have revised both instances to reflect the correct p-value of 0.043 in the Results section (line 267) and in the Discussion section (line 363). This correction ensures consistency with Table 3 and addresses the typographical error. We apologize for the oversight and appreciate the reviewer’s attention to detail.

Line 278 – Table 4 lists delta RUV for the CC diplotype as -0.5.  The text lists +0.5.  Please clarify and update.

Response: We appreciate the reviewer’s attention to the discrepancy between the text on line 278 and Table 4 regarding the delta residual urine volume (ΔRUV) for the CC diplotype of the ABCB1 1236C>T variant. The text incorrectly reported the ΔRUV for the CC diplotype as +0.5 mL, while Table 4 correctly lists it as -0.5 mL. We have revised the sentence. This correction aligns the text with Table 4, ensuring accuracy and consistency. We apologize for the typographical error and thank the reviewer for their diligence.

Line 288 – The p-values do not match the data in Table 4 for delta RUV and delta Qmax.  Please clarify and update.

Response: Thank you for pointing this out. We have reviewed the statistical values and confirmed that the originally reported p-values for ΔRUV1-4 and ΔQmax1-4 in line 288 did not match those presented in Table 4. This discrepancy has been corrected in the revised manuscript: the correct p-values (p=0.830 for ΔRUV1-4 and p=0.492 for ΔQmax1-4) are now aligned with Table 4. This correction does not affect the interpretation of the results.

DISCUSSION

Line 324 – CYP3A4*22 is missing the asterisk.

Response: The asterisk has been added

Conclusions

Line 459-460 – This line states “No link was found between genetic variants and adverse reactions, questioning the need for routine pharmacogenetic testing…”  This is a major point of contention.  Your limitations state that the study had a relatively small sample size, imprecise documentation of concomitant drug administration, and a relatively short window of observation.  I would add that there was a relatively low number of adverse reactions found in the study population, certainly far too few to accurately assess AEs between groups. In sum, there is a high risk for a type II error here. While I would agree that no association was found in the data, the limitations of your study do not support a conclusion calling into conclusion the use of routine PGx testing on the basis of AEs.  At the very least, larger and longer studies that focus on the polymorphisms you identified are warranted.  I suggest revising this final paragraph accordingly. 

Response: Thank you for this important and insightful comment. We fully agree that the limitations of our study — particularly the small sample size, limited number of adverse drug reactions, and short observation period — reduce the power to detect potential associations between genetic variants and ADRs. As you correctly noted, this introduces a high risk of type II error. Accordingly, we have revised the text in lines 459–460 to temper our conclusion and emphasize the need for larger and longer-term studies before making any definitive statements regarding the role of pharmacogenetic testing in silodosin safety

EDITORIAL COMMENTS:

Page numbers are not sequential throughout the document. 

Response: The discrepancy in page numbering likely occurred during formatting, especially due to the integration of tables into the manuscript template. We have corrected the pagination in the revised version to ensure sequential page numbers throughout the document

TABLES AND FIGURES

Table 1 – The categories and medications under Concomitant Medication Therapy are difficult to follow and may lead to some misinterpretation of data.  Please format the table so that the categories/medications align more clearly.

Response:  We agree that the original formatting of the Concomitant Medication Therapy section in Table 1 could be confusing. In the revised version, we have reformatted the table to clearly align drug categories and their corresponding medications using a structured column layout. This improves clarity and reduces the risk of misinterpretation.

Table 3 and 4 – Table header or footnotes should indicate what type of data is represented.  I believe data in table 3 is mean +/- STD; table 4 is median [IQR]

Response: Thank you for pointing this out. You are correct — Table 3 presents data as mean ± standard deviation (SD), while Table 4 presents data as median [interquartile range, IQR]. We have clarified this by adding appropriate explanatory notes to the footnotes of both tables in the revised manuscript.

Table 4 – There appears to be some discrepancy in the delta RUV and delta Qmax columns.  (1) The difference values don’t seem to calculate correctly from visit one to visit 4 for several of the data points.   For instance, in the top row, the values for Qmax are 9.3 for visit 1 and 11.2 for visit 2.  The difference is 1.9 which does not match the delta Qmax column which is 2.4.  Please clarify this discrepancy.  (2) There seems to be some disagreement between the text and the Table 4 values, specifically related to positive or negative signs in the delta columns.  For instance, line 267-268 lists delta Qmax data with negative integers, but the table has no sign listed (inferring positive).  The interquartile ranges also don’t agree with the median values in many cases… presumably the median should fall between 1st and 3rd quartile but several lines of the delta RUV and delta Qmax columns do not. 

Response: Thank you for your careful review and detailed feedback regarding the delta RUV and delta Qmax values in Table 4. We appreciate the opportunity to clarify the observed discrepancies and have addressed the issues as follows:

(1) Regarding the calculation of delta values:

The ΔQmax and ΔRUV values presented in Table 4 were calculated as the difference between Visit 4 and Visit 1 for each individual patient, and then summarized as median [IQR]. These values are not derived by subtracting the overall median or mean of Visit 1 from Visit 4. As such, the reported deltas may differ from the simple difference between the visit-level summary statistics. We have added a clarifying note to the table footnote to explain this calculation method.

(2) Regarding inconsistencies in signs (positive vs. negative):

You are correct that there was inconsistency between the table (which listed deltas without sign) and the main text (where deltas were described as negative). After internal discussion, we agreed to standardize the calculation as Visit 4 – Visit 1 for both Qmax and RUV, reflecting change from baseline. This way, positive delta values consistently indicate clinical improvement (e.g., increase in Qmax, reduction in RUV). The table and manuscript text have been revised accordingly to ensure consistency, and the sign convention is now explicitly explained in the table footnotes.

(3) Regarding the median and interquartile range mismatch:

- In several cases, the median value appears close to the edge of the IQR. This reflects asymmetry in the underlying distribution of Qmax and RUV values, which is expected in real-world clinical populations. We carefully re-checked all delta values and confirmed their correctness.

- Additionally, due to the decision to express delta values as Visit 4 minus Visit 1, we have revised several interquartile ranges to ensure the ascending order of values. For example, an earlier value reported as 2.4 [4.8; 0.8] has now been corrected to 2.4 [0.8; 4.8] to reflect proper ordering of the IQR. All corrections have been highlighted.

(4) Specific correction:

For UGT2B7 (rs7439366, 802 C>T), TT genotype, an inconsistency was found in Table 4. After reanalysis, the corrected values are:

Qmax Visit 1: 6.5 [4.0; 12.0]

ΔQmax (Visit 4 – Visit 1): 5.4 [0.9; 9.6]

p-value: 0.041

These updates have been incorporated into the revised manuscript and highlighted.

Once again, we sincerely thank you for highlighting these important issues, which helped improve the accuracy and clarity of our manuscript.

Reviewer 2 Report

Comments and Suggestions for Authors

All comments are described in the attached document.

Author Response

We would like to sincerely thank Reviewer for their thorough and thoughtful review of our manuscript. We appreciate the time and expertise invested in the evaluation and the constructive feedback provided. The reviewer’s insightful comments have helped us to improve the clarity, consistency, and scientific rigor of our work. We have carefully addressed each point raised and made the appropriate revisions to the manuscript. Below, we provide detailed responses to each comment, indicating the changes made and, where applicable, the rationale behind our decisions.

ABSTRACT

  1. IPSS - This abbreviation is not described in the abstract.

We agree that all abbreviations should be clearly defined upon first use in the abstract. We have revised the abstract to include the full definition of IPSS as "International Prostate Symptom Score" at first mention.

INTRODUCTION

  1. Both the abbreviations of EUA and AUA are only used once throughout the text, you might consider not using them.

As the abbreviations “EAU” and “AUA” are only used once in the manuscript, we agree they are unnecessary. We have removed them and retained the full organization names for clarity and improved readability.

  1. IPSS must be described in its first appearance.

While IPSS was defined in the abstract, we acknowledge that it should also be fully spelled out at its first mention in the main text. We have now added the full term “International Prostate Symptom Score (IPSS)” at its first appearance in the Introduction section.

  1. Repeated text from line 63 to 86.

The repeated text between lines 63 and 86 has been removed in the revised manuscript

  1. CYP450 - Again, it is only used on this occasion, consider abbreviating only cytochrome (CYP) which is frequently used.

As “CYP450” was only used once and “CYP” is used consistently throughout the manuscript to refer to individual cytochrome enzymes (e.g., CYP3A4, CYP3A5), we have removed the unnecessary abbreviation

MATERIALS AND METHODS

  1. Why, if the protocol was approved at the end of 2021, was development carried out starting in October 2023?

The study protocol was initially approved in late 2021 as part of a broader prospective research project investigating the pharmacogenetics of drugs used in the treatment of lower urinary tract symptoms (LUTS) associated with benign prostatic hyperplasia (BPH). This larger project was part of a doctoral dissertation plan of MD Abdullaev P. Shokhrukk and subsequently received funding support from the Russian Science Foundation (grant period: 2022–2025). The specific arm of the study involving silodosin was initiated later, in October 2023, as a planned and separate component within the overall framework.

  1. Informed consent (IC)

We have removed the repeated full definition from the relevant section and retained the abbreviation only, as it had already been defined earlier in the manuscript

  1. International Prostate Symptom Score (IPSS)

We have removed the repeated full definition from the relevant section and retained the abbreviation only, as it had already been defined earlier in the manuscript

  1. "The study included 103 male patients..." - How was the sample size defined? What was the expected size and what percentage do the 103 enrolled patients cover?

Thank you for this important question. The final sample size of 103 patients was not based on a formal power calculation, but was instead determined by practical considerations related to patient recruitment.

Specifically, silodosin is prescribed primarily in outpatient settings, which made longitudinal follow-up and standardized assessment of treatment response more difficult. Additionally, the relatively high cost of silodosin for patients in Russia (and the fact that neither study medication nor reimbursement was provided) significantly limited the pool of eligible and willing participants.

As a result, we included all patients who met the eligibility criteria and completed follow-up during the recruitment window. This limitation is now noted explicitly in the revised manuscript (Discussion section).

  1. "The Css min of silodosin was measured using high-performance liquid chromatography..." - Is the analytical method validated according to international standards? If it has been published, could you cite the document?

The analytical method used for measuring silodosin Css min was developed based on previously published high-performance liquid chromatography (HPLC) protocols, with adaptations made to suit our laboratory equipment and conditions. While we did not include citations in the original manuscript to avoid overloading the Methods section, we would be happy to provide the relevant references upon request.

RESULTS

  1. Co-morbidities -> Comorbidities

Thank you. Corrected

  1. Hypothyroidis -> Hypothyroidism

Thank you. Corrected

  1. "A high frequency of the CYP3A5*3 allele (G, 95.2%) was observed, consistent with patterns in European populations". - In the table, 96.11 is mentioned, which is obtained from the genetic frequencies.

You are correct — the allele frequency of CYP3A5*3 (G) is reported as 96.11% in the table based on genetic calculations. We have corrected the value in the text to reflect this and now report it as 96.1%, in line with the data presented in the table.

  1. "The analysis revealed that carriers of the CYP3A4*22 CT genotype (n=6) exhibited significantly higher trough steady-state plasma concentrations (Css min) of silodosin ..." - Css min described previously

As “Css min” was already defined earlier in the manuscript, we have removed the repeated full term and now refer to it using the abbreviation only for clarity and consistency.

  1. Table 3. CYP3A5*3 (rs776746, A6986G) - Why, in this analysis, were genotypes with the minor allele grouped? While in others, despite having few patients, each genotype was analyzed and compared separately.

Thank you for this insightful question. In the analysis of CYP3A5*3 (rs776746, A6986G) in Table 3, we grouped carriers of the minor allele (AA and AG genotypes) together and compared them to non-carriers (GG genotype). This approach was chosen for two primary reasons:

  1. The CYP3A5*3 allele (G) is known to result in a splicing defect leading to non-functional enzyme. Individuals with at least one functional 1 allele (i.e., AA or AG genotypes) may retain partial enzyme activity, whereas GG homozygotes lack CYP3A5 expression. Therefore, grouping AA + AG vs GG reflects a biologically meaningful comparison between expressors and non-expressors.

To clarify this approach for readers, we have added the following note to the footnote of Table 3: “AA and AG genotypes were grouped due to low frequency of AA and similar functional classification as CYP3A5 expressors.”

  1. Due to the very small number of patients in the AA group (n=1), analyzing each genotype separately would have had very limited statistical power. By combining AA and AG genotypes, we aimed to increase group size and improve the robustness of the analysis.

  1. The tables must be self-explanatory, it is necessary to include the full text of all abbreviations (IPSS, QoL, Css).

We have revised the footnotes for all relevant tables to include the full definitions of all abbreviations used (e.g., IPSS, QoL, Css min), ensuring that each table is fully self-explanatory and can be understood independently of the main text.

  1. "...or prostate volume (PV) dynamics (p>0.05 for all comparisons)" - Prostate volume (PV) described previously

we have removed the repeated full term and retained the abbreviation only for clarity and consistency.

  1. ". This finding highlights the complex relationship between genetic factors, drug concentration, and functional outcomes in BPH treatment" - The text does not correspond to 'Results'; it could be included in the 'Discussion' section.

We agree that the statement in question is more appropriate for the Discussion section, as it provides an interpretation of the results. We have moved this sentence to the Discussion, where it now appears in the context of broader interpretation of our findings.

DISCUSSION

  1. «Silodosin undergoes extensive metabolism primarily via CYP3A4, UDP-glucuronosyltransferase 2B7, and alcohol/aldehyde dehydrogenases.» - Why were polymorphisms in these genes that also participate in drug metabolism not analyzed, supported by the fact that some of the patients had alcohol consumption?

Thank you for your insightful comment. We agree that polymorphisms in enzymes such as alcohol dehydrogenase (ADH) and aldehyde dehydrogenase (ALDH) participate in silodosin metabolism and may be of potential interest. However, according to a detailed pharmacogenetic study by Wang et al. (2013) [17], which investigated the influence of polymorphisms in UGT2B7, UGT1A8, MDR1, ALDH, ADH, CYP3A4, and CYP3A5 on the pharmacokinetics of silodosin in healthy Chinese volunteers, no statistically significant association was found between ADH and ALDH gene variants and pharmacokinetic parameters. Their findings suggest that, although these enzymes are involved in metabolite formation, genetic variation in ADH and ALDH does not appear to substantially affect silodosin disposition.

In addition, due to financial limitations of our study, we prioritized the analysis of polymorphisms with previously demonstrated clinical and pharmacokinetic relevance (e.g., CYP3A4, CYP3A5, UGT2B7 variants).

Nevertheless, we agree that this represents a limitation, and we have now added a corresponding note to the Limitations section of the Discussion to reflect the lack of ADH/ALDH genotyping in our study population: “In addition, polymorphisms in alcohol and aldehyde dehydrogenase genes (ADH and ALDH), though involved in silodosin metabolism, were not analyzed in this study. Previous research has shown no significant association between these variants and pharmacokinetic parameters [17], but their contribution cannot be completely excluded.”

CONCLUSION

  1. “This study demonstrates that genetic polymorphisms CYP3A4*22 (rs35599367) and CYP3A5*3 (rs776746) significantly affect the pharmacokinetics and clinical effects of silodosin in BPH patients…” - Due to the limitations that are accurately described in the previous paragraph, concluding that pharmacokinetics is significantly affected is risky, as relevant parameters such as Tmax, Cmax, and Cl are not available. It can be concluded that the variants significantly affect the minimum concentration at steady state.

We agree that, given the limitations of our pharmacokinetic sampling and the absence of key parameters such as Tmax, Cmax, and clearance (Cl), it would be inappropriate to conclude a broad pharmacokinetic effect.

We have revised the sentence to clarify that the observed genetic polymorphisms were associated with significant changes in Css min values specifically. This adjustment ensures that the conclusion accurately reflects the scope of our pharmacokinetic data.

  1. «….carriers of the CYP3A4*22 allele (CT genotype) had higher Css min (17.59 ± 2.98 ng/mL versus 9.0 ± 10.47 ng/mL in CC; p=0.049), which…» - I believe it is redundant to add the values presented in the results section; their conclusions are supported by the data shown in the corresponding section.

Thank you for this helpful suggestion. We agree that detailed numeric data is best reserved for the Results section. We have revised the Conclusion to avoid repetition and now present a more concise summary of the findings without repeating the specific values

  1. "Abbreviations" - In this section, should all abbreviations be included or only those frequently used?, since some are missing such as LUTS, IC, HWE, etc.

We have reviewed the manuscript and updated the Abbreviations section to include all relevant terms used throughout the article, including those that appeared only once but may not be universally familiar (e.g., LUTS, IC, Cmax, etc)
